# $S^6$-DAMON: Bridging Self-Supervised Speech Models and Real-time Speech Recognition

## Abstract

There has been an growing demand for deep neural network (DNN) powered automatic speech recognition (ASR) on mobile platforms for real-time speech recognition. However, ubiquitous on-device ASR systems are still hindered by two bottlenecks: (1) the lack of large-scale transcribed speech data especially for low-resource spoken languages and (2) the large gap between DNNs' prohibitive complexity and mobiles' limited resources. In parallel, speech models pretrained via self-supervised learning (SSL) have emerged to reduce the reliance on the availability of transcribed speech data, which however further enlarges the efficiency gap because they often adopt large transformers to ensure expressive speech representations. Thus, it is highly desired to trim down the complexity of speech SSL models to enable real-time on-device ASR. This is particularly challenging since only structured sparsity can favor hardware efficiency in commercial devices, under which the speech representation learned by SSL could easily be demolished. To this end, we develop a framework dubbed $S^6$-DAMON to pursue **s**tructured **s**parsity in **s**peech **SS**L models via **da**ta-**mo**del co-compressio**n**. On the data side, leveraging both the duration of each phoneme and the pauses between the words/phonemes of human utterances, we propose a **sal**ient **a**udio token **d**etector, dubbed SALAD, to remove input audio tokens that are redundant; On the model side, we identify that the failure of the SOTA ASR pruning method under structured sparsity is caused by the sparsity discrepancy between finetuning/deployment and their limited learnability of sparsity distributions, and then tackle it via a new ASR pruning pipeline dubbed SAFARI, which adopts a three-step pipeline - **s**p**a**rsify, **f**inetune, and **a**djust sp**a**r**si**ty. Extensive experiments validate that $S^6$-DAMON can enable real-time ASR with limited transcribed speech data requirements while maintaining decent recognition performance. All source codes will be released upon acceptance.

## 1 Introduction

Recent breakthroughs in deep neural networks (DNNs) have tremendously advanced the field of Automatic Speech Recognition (ASR), enabling record-breaking end-to-end ASR systems (Hannun et al., 2014; Chan et al., 2016; Zhang et al., 2020; Gulati et al., 2020). Considering that speech is one of the basic input modalities from users of intelligent mobile devices, there has been an increasing interest in the development and deployment of on-device ASR systems. For example, intelligent assistants (Meta-AI, 2022; Vox, 2022) are highly desired in next-generation augmented reality and virtual reality (AR/VR) devices for enabling immersive AR/VR experiences. This has called for advanced speech technologies in order to deliver accurate and real-time ASR systems.

There still remain two critical efficiency bottlenecks for ubiquitous on-device ASR systems, including (1) *data efficiency:* big data is often not practical for ASR since collecting transcription on a large scale is costly or not even possible, especially for low-resource spoken languages and (2) *model efficiency:* the often limited on-device resources stand at odds with the complexity of deep ASR models, making it particularly challenging to satisfy real-time ASR requirements. To promote the aforementioned data efficiency, recent advances in self-supervised learning (SSL) for speech representation (Chi et al., 2020; Baevski et al., 2020; 2022) have demonstrated empirical success and become the de-facto paradigm for low-resource ASR, where SSL models pretrained on unlabeled audio data can be generalized to handle low-resource transcribed speech after being finetuned. However, this could further aggravate the model efficiency bottleneck as giant transformers (Vaswani

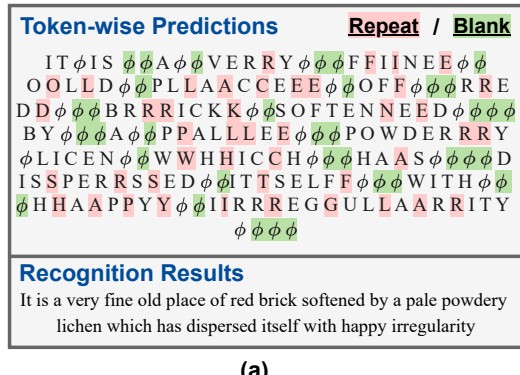
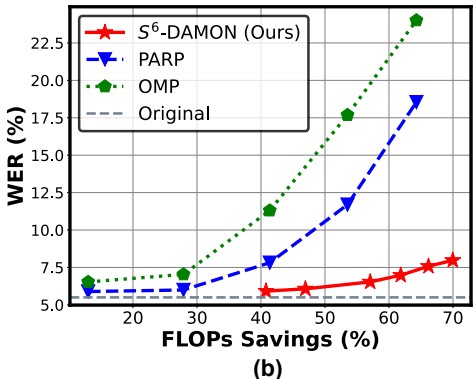

**(a)**                                 **(b)**

Figure 1: (a) An example from LibriSpeech for illustrating two types of non-salient audio tokens; (b) The trade-offs between WER on LibriSpeech test-clean and FLOPs savings achieved by different ASR compression schemes on top of wav2vec2-base finetuned on LibriSpeech-100h.

et al., 2017) (e.g., > 90M parameters) are often adopted in state-of-the-art (SOTA) speech SSL models to ensure effective representation learning during SSL, making it increasingly more challenging for on-device deployment. Therefore, it is imperative to compress speech SSL models while maintaining their generalizable speech representation for delivering efficient ASR systems.

Despite the demanding need, it is non-trivial to narrow the gap between large speech SSL models and constrained resources in mobile devices. First, under the SOTA pretrain-then-finetune paradigm, most useful features are learned during the SSL stage and then pretrained speech SSL models only slightly adapt their weights to encode task-specific information during finetuning, whereas it is difficult to learn a sparsity distribution during finetuning while maintaining the fidelity of the speech representation *given the low-resource downstream speech*. Note that this is particularly challenging for ASR due to the more stringent low-resource settings, e.g., LibriSpeech-10m (Panayotov et al., 2015) for ASR only contains 48 sentences for training and development, whereas the CoLA dataset (Wang et al., 2018) for natural language processing (NLP) contains 9594 sentences. Second, only structured sparsity can favor hardware efficiency in commercial mobile devices, which however will pose a more severe destruction during finetuning on the SSL speech representation learned during pretraining than unstructured sparsity, e.g., enforcing structured sparsity in the SOTA unstructured ASR pruning framework called PARP (Lai et al., 2021) will cause a >8% increase in word-error-rate (WER) under only a 20% sparsity on wav2vec2-base/LibriSpeech-1h. Third, considering that ASR corresponds to a sequence-to-sequence task where the alignment between inputs and outputs is monotonic, the compression process is thus required to be more meticulous in preserving the information of useful audio frames than compressing classification-task models.

**Our Contributions.** We develop a framework dubbed S[6]-DAMON which is **the first** to pursue **s**tructured **s**parsity in **s**peech **SSL** models under low-resource settings via **da**ta-**mo**del co-compressio**n** for enabling real-time on-device speech recognition.

**On the data side**, S[6]-DAMON leverages the intrinsic redundancy in human speech. As the duration of each phoneme and the pauses between the words/phonemes of human utterances, the sampled audio frames and the corresponding extracted audio tokens, i.e., inputs for the transformers, may (1) repeat the previous tokens, or (2) stand as blank, contributing little to the final recognition (see an example in Fig. 1 (a)). We call both as *non-salient audio tokens (NATs)* and the first-appearing tokens that are indispensable for ensuring monotonic recognition as *salient audio tokens (SATs)*. Properly removing NATs can lead to non-trivial savings in model efficiency while better maintaining the accuracy than removing SATs, e.g., NATs account for 50.6% of the total tokens on LibriSpeech test-clean based on token-wise annotations from finetuned wav2vec2-base. As only sentence-level transcriptions are annotated in ASR datasets and token-wise labels are not available to classify SATs/NATs, we design a **sal**ient **a**udio token **d**etector called SALAD and train it in a semi-supervised manner based on the pseudo token-wise labels annotated by finetuned speech SSL models on untranscribed speech. A high recall is enforced to ensure the coverage of SATs, thus properly removing NATs detected by SALAD in inference can better maintain the speech representation fidelity. **On the model side**, we discover that the failures of SOTA ASR pruning method PARP (Lai et al., 2021) under structured sparsity are caused by (1) the sparsity discrepancy between finetuning/deployment, i.e., PARP finds the

necessity of maintaining the flexibility of sparsity distributions and thus keeps all weights updatable, including pruned ones, during finetuning, resulting in a discrepancy against the hard-pruned weights during deployment. The mismatched finetuning/deployment process could demolish the learned speech representations especially under structured sparsity; (2) the limited learnability of sparsity distributions in the PARP pipeline due to the intrinsically low learning rates during finetuning, leaving the space of sparsity masks significantly under-explored. We thus propose a new ASP pruning pipeline dubbed SAFARI (i.e., **spa**rsify, **f**inetune, and **a**djust spa**r**s**i**ty) to strictly zero-out the pruned neurons during finetuning for minimizing the finetuning/deployment discrepancy, followed by a sparsity adjustment step to adaptively evolve the sparsity masks and thus ensure the learnability of sparsity distributions. We summarize our contributions as follows:

- We propose a data-model co-compression framework, dubbed $S^6$-DAMON, which for the first time chases structured sparsity in both input audio tokens and model structures of speech SSL models to empower real-time on-device ASR under a low-resource setting;

- We develop a semi-supervised method for training a lightweight module dubbed SALAD to distinguish SATs/NATs for the purpose of structurally removing redundant audio tokens;

- We identify the underlying causes for the failures of the SOTA ASR pruning method and then develop an ASR pruning pipeline dubbed SAFARI to enable high structured sparsity in speech SSL models while maximally maintaining the ASR accuracy;

- Experiments show that as compared to the SOTA ASR pruning method PARP, our $S^6$-DAMON can (1) achieve a $1.96\times$ speed-up on Pixel 3 mobile phone with an absolute 2.49% WER reduction, and (2) win an absolute 10.96% lower WER for reducing >64% FLOPs of wav2vec2-base as shown in Fig. 1 (b), indicating that our method has taken an important step towards bridging speech SSL models and real-time speech recognition.

## 2 RELATED WORK

**Automatic speech recognition.** Early ASR systems (Bridle, 1990; Sha & Saul, 2006; Tang, 2009; Jaitly et al., 2012; Mao et al., 2019; Adams & Beling, 2019) mainly build on top of the combinations of hidden Markov models with Gaussian mixture models or DNNs, and often integrate multiple modules, e.g., an acoustic model, a language model, and a lexicon model, that are separately trained. Driven by recent advances in DNN structures, diverse end-to-end ASR systems have been proposed, which process raw audio inputs in an end-to-end manner, including CTC (Graves et al., 2006)-based models (Hannun et al., 2014; Amodei et al., 2016; Graves & Jaitly, 2014; Miao et al., 2015; Eyben et al., 2009), recurrent neural network(RNN)-transducers (Graves, 2012; Graves et al., 2013; Rao et al., 2017; Dong et al., 2018b), and sequence-to-sequence models (Chorowski et al., 2015; Bahdanau et al., 2016; Chan et al., 2016; Zhang et al., 2017; Chiu et al., 2018; Prabhavalkar et al., 2018). Specifically, from the model structure perspective, transformer-based models (Zhang et al., 2020; Gulati et al., 2020; Dong et al., 2018a; Wang et al., 2020) have been widely adopted thanks to their superior expressiveness and capabilities for modeling long-range dependencies.

**Self-supervised learning for speech representation.** Due to the high cost and/or impracticality of collecting large-scale speech data, the pretrain-then-finetune paradigm has become increasingly popular for empowering low-resource ASR. To learn rich speech representation via SSL, early works design generative models for inferring the latent variables of speech units (Hsu et al., 2017a; van den Oord et al., 2017; Hsu et al., 2017b; Ebbers et al., 2017; Khurana et al., 2020). Recently, prediction-based SSL methods have gained more attention, where the models are trained to reconstruct the contents of unseen frames (Chung et al., 2019; Liu et al., 2020; Chi et al., 2020; Ling & Liu, 2020; Baevski et al., 2022) or contrast the features of masked frames with those of randomly sampled ones (Oord et al., 2018; Baevski et al., 2020; Conneau et al., 2020; Baevski et al., 2019a; Hsu et al., 2021). In parallel, some works combine both predictive and contrastive objectives (Baevski et al., 2019b;a) or integrate contrastive learning and masked language modeling (Chung et al., 2021; Bapna et al., 2022). We refer the readers to the survey (Liu et al., 2022) for more details. However, SOTA speech SSL models (Baevski et al., 2020; Hsu et al., 2021; Baevski et al., 2022; Conneau et al., 2020; Chiu et al., 2022) often adopt large transformers for ensuring effective representation learning during SSL and require prohibitive model sizes, making it difficult to achieve real-time speech recognition on mobile devices.

**ASR pruning.** To compress large-scale ASR models while maintaining their generalizable representation, ASR pruning has gained a growing attention. Early works prune either the decoding

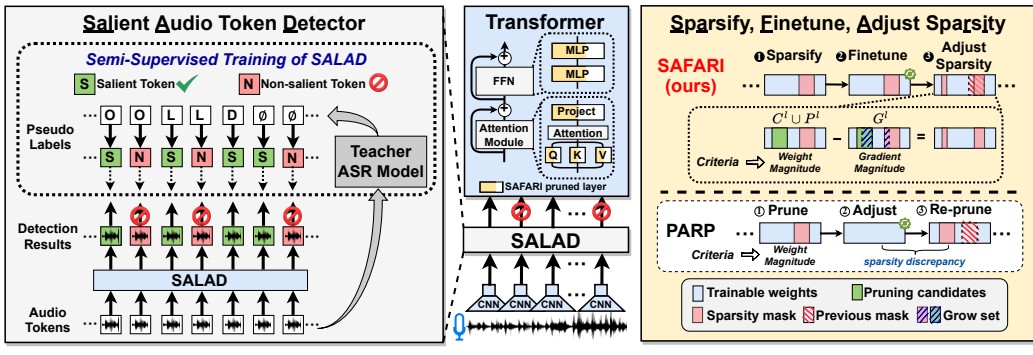

Figure 2: An overview of our proposed $S^6$-DAMON, integrating two enabling components called (a) SALAD and (b) SAFARI for chasing structured sparsity in input data and model structures, respectively (zoom-in for better view).

search space (Abdou & Scordilis, 2004; Pylkkönen, 2005; Xu et al., 2018; Zhang et al., 2021) or the HMM state space (Van Hamme & Van Aelten, 1996). Recent works have shifted its focus to pruning end-to-end ASR models (Venkatesh et al., 2021; Shi et al., 2021; Li et al., 2021b; Braun & Liu, 2019; Gao et al., 2020; Shangguan et al., 2019; Wu et al., 2021; Ding et al., 2021). Recently, (Lai et al., 2021; Prasad et al., 2022; Zhao et al., 2021) prune speech SSL models towards more efficient low-resource ASR, however, they all adopt unstructured pruning which barely favors hardware efficiency in commercial devices. (Lee et al., 2022; Chang et al., 2022) distill the knowledge of pretrained speech SSL models to lightweight student models, but require human expertise to manually design the student model, which can be impractical without utilizing the properties of speech signals, causing inferior ASR accuracy. In contrast, $S^6$-DAMON learns to *automatically* and *structurally* prunes the redundancy of speech SSL models, achieving a triple-win in data, model, and labor efficiency.

## 3 THE PROPOSED $S^6$-DAMON FRAMEWORK

### 3.1 $S^6$-DAMON: FRAMEWORK OVERVIEW

**Rationale.** Current commercial mobile devices can only benefit from structured sparsity, whereas most useful features of speech SSL models are learned during the SSL stage, which could be easily demolished when enforcing structured sparsity, especially when being finetuned under a low-resource setting. To this end, our $S^6$-DAMON tackles this challenge based on a twofold rational: (1) instead of compressing only one dimension, $S^6$-DAMON exploits the redundancy in both the input audio tokens and model structure; and (2) to preserve the monotonic alignment between the input audio and output transcriptions, $S^6$-DAMON identifies and then tackles the key bottlenecks for reducing the complexity of both the input and model while maintaining the fidelity of speech representation.

**Overview.** As shown in Fig. 2, $S^6$-DAMON performs data-model co-compression via (1) a SALAD module which detects and skips redundant tokens, i.e., NATs as introduced in Sec. 1, and (2) a SAFARI pipeline which pursues structured model sparsity. For compressing a given speech SSL model, $S^6$-DAMON features a three-stage pipeline: ❶ finetune the speech SSL model that serves as a teacher model on the low-resource transcribed speech; ❷ train SALAD in a semi-supervised manner based on the annotations of the teacher model (see Sec. 3.2); and ❸ perform a joint optimization of input data sparsity and model sparsity on top of speech SSL models based on the SAFARI pipeline with a portion of detected NATs removed (see Sec. 3.3 and Sec. 3.4).

### 3.2 $S^6$-DAMON: CHASE STRUCTURED SPARSITY IN INPUT DATA

Here we introduce the deign and training of our proposed SALAD module, which is to detect and skip redundant audio tokens, i.e., NATs. Note that although (Kim et al., 2021; Wang et al., 2021) have attempted to prune transformers' tokens in a layer-wise manner in the NLP domain, they do not consider the intrinsic properties of human speech and such a layer-wise dynamic skipping workload can barely favor hardware efficiency in commercial devices. In contrast, SALAD explicitly exploits the redundancy in repeated/blank regions of human utterances, favoring efficiency in commercial devices, while maintaining the monotonic alignment between input speech and output transcriptions, a unique property in the speech domain, by enforcing a high recall on SATs as elaborated below.

**SALAD's input and structure.** SOTA speech SSL models (Baevski et al., 2020; Hsu et al., 2021; Baevski et al., 2022) sample and convert raw audio frames into audio tokens via convolutional feature extractors, which are then processed by a transformer backbone to generate corresponding contextualized representation. As the convolutional feature extractors are often fixed after SSL pretraining to ensure effective audio feature extraction (Baevski et al., 2020), SALAD is applied on the extracted audio tokens after the convolutional feature extractor, and classifies each audio token as a SAT/NAT. This ensures that all transformer layers can benefit from reducing the same number of audio tokens, favoring efficiency in commercial devices. Specifically, SALAD consists of four lightweight convolutional layers and outputs a binary classification between SAT and NATs. Its structure is provided in the appendix, which only contains 0.35M parameters and accounts for <0.4% floating-point-operations (FLOPs) of the original transformer.

**SALAD's semi-supervised learning pipeline.** Considering the lack of token-wise ground truth, we train $SALAD(\cdot; \theta_S)$ in a semi-supervised manner, i.e., using a finetuned ASR model to provide pseudo labels for each token on untranscribed speech (see Fig. 2 (a)). Specifically, given a speech SSL model, we first finetune it on the available transcribed speech (e.g., LibriSpeech-1h) to create a teacher model $M_T(\cdot; \theta_T)$, which is then used to annotate a larger amount of untranscribed speech (e.g., LibriSpeech-10h) to acquire the pseudo character/phoneme labels for each audio token. Correspondingly, the pseudo binary labels of SATs/NATs can be derived for each audio token based on whether it repeats the previous token or is a blank token, serving as the training signals for SALAD.

Enforce a high recall on SATs. One issue is that the consequence of classifying a SAT to a NAT is severe, under which SATs are more likely to be mistakenly skipped in inference, harming the monotonic input/output alignment. It is thus highly desired to maximize the coverage of SATs (i.e., a high recall on SATs) during training SALAD. We exert a larger penalty when a SAT is misclassified as a NAT to simply achieve this. The training process of SALAD can be formulated as:

$$\arg\max_{\theta_S} \sum_{i=1}^{t} \alpha_i L(SALAD(x_i; \theta_S), Bin(M_T(x_i; \theta_T))) \tag{1}$$

where $L$ is a binary cross entropy loss, $x_i$ is the $i$-th audio token extracted by the convolutional feature extractor, $Bin$ denotes a transformation from pseudo character/phoneme labels to binary labels of SATs/NATs, and $\alpha_i$ is a penalty coefficient for enforcing high recalls on SATs.

**Speech SSL model finetuning with SALAD.** Since skipping audio tokens would change the speed and rhythm of an input speech, a finetuning process is required to fill in the domain gap. Specifically, we finetune the target speech SSL model integrated with SALAD, where a certain ratio of NATs detected by SALAD is removed before being fed into the transformer. Intuitively, although NATs are less likely to impact the monotonic input/output alignment as compared to SATs, removing NATs still results in domain gaps in terms of speech speed and rhythm between pretraining and finetuning.

Implementation. To balance ASR accuracy and efficiency, we set a skip ratio $sr$ for NATs in all the input audio when finetuning with SALAD, i.e., for the NATs detected by SALAD in an input audio clip, we remove the top $sr$ sorted in terms of confidence score predicted by SALAD, and the remaining NATs and all detected SATs are then fed into the transformer. Since different audio clips contain different percentages of NATs, which can cause different audio token lengths for samples in a batch, we pad each sample to the largest token length of each batch during finetuning.

### 3.3 $S^6$-DAMON: Chase Structured Sparsity in Model Structures

**Rethink the SOTA ASR pruning method.** The key spirit of PARP (Lai et al., 2021) is to adjust the sparsity mask so that the pruned weights are learnable via gradients and thus the sparsity distributions can be updated during finetuning, which is crucial for pruning speech SSL models as the weight magnitudes inherited from SSL pretraining may not accurately indicate the importance of neurons for downstream tasks. The importance of such learnable sparsity distributions is validated in PARP with *unstructured* pruning against one-shot/iterative magnitude pruning (OMP/IMP) (Lai et al., 2021).

**Identified issues of the SOTA ASR pruning method.** Directly extending PARP to a structured pruning setting will cause a failure. We extend PARP's setting to structured sparsity, i.e., remove all connections from the pruned input neurons, for pruning wav2vec2-base on LibriSpeech-1h under a 20% sparsity ratio and vary the pruning intervals in terms of iterations between prune/re-prune. From Tab. 1, we can see that (1) the original pruning interval adopted by PARP (i.e., 50 iterations) leads to

an absolute 23.93% WER increase over standard wav2vec2-base (18.96% WER); (2) setting a small pruning interval could lead to reduced WER over OMP; and (3) larger pruning intervals consistently cause more notable performance degradation, where the softly pruned weights diverge more from zero as they can be updated in PARP's adjustment step.

Analysis. This set of experiments indicates that although enhancing the sparsity masks' learnability in PARP is beneficial, the resulting sparsity discrepancy between finetuning (i.e., not exactly zero) and final deployment (i.e., hard pruning) can demolish the delicated speech representation inherited from SSL pretraining, es-

Table 1: Apply OMP and PARP with different pruning intervals for compressing wav2vec2-base on LibriSpeech-1h under 20% sparsity.

| Method | Pruning Interval | | | | | Learn -ability | No Dis -crepancy |
|---|---|---|---|---|---|---|---|
| | 1 | 2 | 5 | 10 | 50 | | |
| PARP | **26.90** | 27.30 | 31.18 | 32.77 | 42.89 | ✔ | ✗ |
| OMP | | | 29.01 | | | ✗ | ✔ |

pecially under structured sparsity. *Additionally*, the flexibility of adjusting the sparsity masks in PARP is still limited, e.g., only <3% elements in the sparsity masks is updated throughout the finetuning process, which aligns with PARP's observed >99% IOU between the initial/final subnetworks. This is caused by the intrinsically low learning rates during finetuning, thus updating the sparsity distributions via gradients in PARP leaves the potential of sparsity distributions largely under-explored.

**The SAFARI pipeline.** The above analysis indicates that the key to pursue structured sparsity in speech SSL models is to *ensure the learnability of sparsity distributions while minimizing the sparsity discrepancy between finetuning and deployment*. Thus, we propose a new pruning pipeline SAFARI: ❶ *sparsify*: sparsifies a given speech SSL model to the target sparsity ratio $sp$ based on the weight magnitudes; ❷ *finetune*: finetune the model weights with the sparsity mask applied, i.e., the pruned weights are zero-outed without receiving gradients to avoid the sparsity discrepancy; and ❸ *adjust sparsity*: the sparsity mask is adaptively adjusted for boosting the learnability of sparsity distributions. Steps ❷ and ❸ are iterated towards convergence. SAFARI can be viewed as an intermediate choice between OMP and PARP, marrying the former's stability and the latter's learnability.

Implementation of SAFARI. There can be different ways to implement the above spirit, i.e., SAFARI. Inspired by (Evci et al., 2020), we adopt gradient magnitudes as a criterion to adjust the sparsity masks in a prune-and-grow manner (see Fig. 2 (c)). Specifically, in each sparsity adjustment step, for a set of neurons $|N^l|$ and a set of pruned neurons $P^l$ ($|P^l|/|N^l| = sp$) in the $l$-th layer, SAFARI ① selects $ar$ neurons from $N^l \setminus P^l$ as the pruning candidate set $C^l$ ($|C^l|/|N^l| = ar$) based on a pruning criterion, where $ar$ is a predefined adjustment ratio, and ② chooses $ar$ neurons from the joint set of pruning candidates and pruned neurons $P^l \cup C^l$ ($= P^l + C^l$) to form a grow set $G^l$ based on a grow criterion, which are allowed to be updated by the gradients of the next finetuning step. Therefore, the new sparsity mask applied in the next finetuning step is built by $P^l + C^l - G^l$, which has a constant sparsity ratio $sp$. More specifically, we adopt the $\ell_1$-norm of the weight vectors from a neuron, i.e., $||W_{i,\cdot}^l||_{\ell_1}$ for the $i$-th neuron, as the pruning criterion (i.e., prune the smallest ones), and the corresponding gradients $||\frac{\partial L}{\partial W_{i,\cdot}^l}||_{\ell_1}$ as the grow criterion (i.e., grow the largest ones).

## 3.4 $S^6$-DAMON: JOINT DATA-MODEL CO-COMPRESSION

To perform joint optimization of input data sparsity and model sparsity, we integrate the target speech SSL model with SALAD and finetune it via the SAFARI pipeline with a portion of detected NATs removed. To push forward the achievable accuracy-efficiency trade-off, $S^6$-DAMON can optionally enable a semi-supervised distillation mechanism to boost the ASR accuracy, especially under a large compression ratio. Specifically, we distill the knowledge of the teacher model mentioned in Sec. 3.2 to the compressed model in a layer-wise manner during finetuning on top of a mixed dataset composed of both transcribed speech and untranscribed speech, where the pseudo labels on untranscribed speech are annotated by the teacher model. Note that the teacher model is only finetuned on the limited transcribed speech. The semi-supervised distillation process can be formulated as:

$$\arg\max_{\theta} \sum_{x \in D_T \cup D_U} \sum_{l=1}^{L} MSE(h_\theta^l(x), h_{\theta_T}^l(x)) + \sum_{x \in D_T} CTC(h_\theta^L(x), y) \qquad (2)$$

where $D_T$ and $D_U$ are the transcribed/untranscribed speech, respectively, $MSE$ and $CTC$ are the loss functions, $h_\theta^l(x) = M^l(SALAD(x; \theta_S); \theta)$ is the hidden representation for the remained tokens in the $l$-th layer of the compressed model $M$, $h_{\theta_T}^l(x) = \mathbb{S} \circ M_T^l(x; \theta_T)$ is the corresponding hidden representation of the teacher model and $\mathbb{S}$ is a selection operator for only calculating the MSE loss on the remained tokens determined by SALAD.

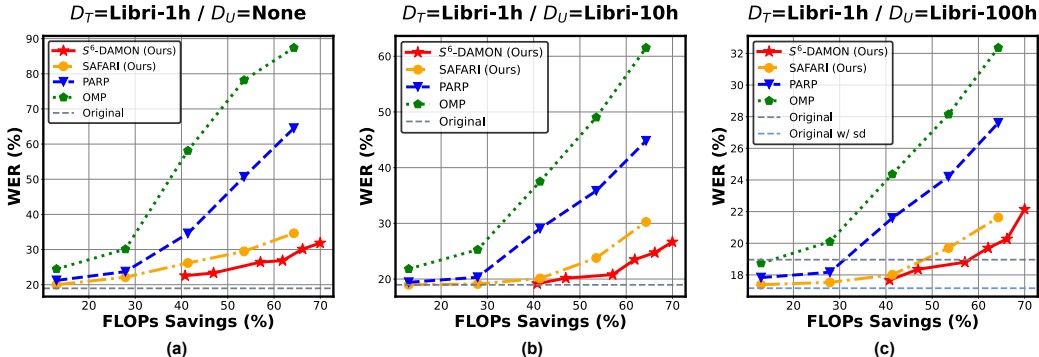

Figure 3: Benchmark our S$^6$-DAMON and SAFARI with SOTA ASR pruning methods PARP and OMP on wav2vec2-base with transcribed LibriSpeech-1h and different untranscribed resources. "w/ sd" denotes applying the semi-supervised distillation for finetuning the original model, which could notably reduce the WER when the untranscribed LibriSpeech-100h is available.

## 4 EXPERIMENTAL RESULTS

### 4.1 EXPERIMENT SETUP

**Models and datasets.** We evaluate our S$^6$-DAMON on four SOTA speech SSL models, including wav2vec2-base/large (Baevski et al., 2020), data2vec (Baevski et al., 2022), and hubert (Hsu et al., 2021) pretrained on LibriSpeech-960h (Panayotov et al., 2015). We evaluate the compression effectiveness under different resource settings, including ASR on LibriSpeech-1h/10h/100h following the split in (Baevski et al., 2020), and phoneme recognition (PR) on different spoken languages from CommonVoice (Ardila et al., 2019) with 1h transcribed speech per language, following the split in (Conneau et al., 2020). In addition to ASR, we also consider six speech processing tasks from SUPERB (Yang et al., 2021). For results on LibriSpeech, we report the WER on test-clean by default.

**Finetuning settings:** We implement S$^6$-DAMON on top of fairseq (Ott et al., 2019) and we follow the default finetuning settings for each task as elaborated in Appendix. B.

**S$^6$-DAMON settings:** For SALAD training, we adopt the same training schedule as finetuning the speech SSL model weights and the $\alpha_i$ in Eq. 1 is 10 for penalizing mistakes on SATs otherwise 1. For SAFARI, we adjust the sparsity every 50 iterations and fix the sparsity mask after 10k iterations for all experiments. By default, the adjustment ratio $ar$ is set the same as the target sparsity $sp$ if not specifically stated, which is justified by the ablation study in Appendix. A.2. We adopt a NAT skip ratio $sr$ of 0.4/0.6/0.8 (i.e., $sr$ of NATs are removed) and note that the final ratio of skipped tokens to the total tokens would depend on the amount of NATs detected in the given speech.

### 4.2 BENCHMARK WITH SOTA ASR PRUNING METHODS

Considering feed forward networks (FFNs) are more sensitive to structured pruning than self-attention (SA) as observed in Appendix. A.1 thus for both our method and baselines, given a target sparsity $sp$, we by default set their sparsity to satisfy $(sp_{SA} + sp_{FFN})/2 = sp$ and $sp_{SA} - sp_{FFN} = 0.2$, which achieves better ASR accurary with a comparable FLOPs as compared to uniformly setting a sparsity of $sp$. For PARP, we adopt its best-performed setting for structured pruning, i.e., update the sparsity every one iteration for maximally avoiding the sparsity discrepancy issue.

**Benchmark on English ASR under different low-resource settings.** We benchmark our S$^6$-DAMON with OMP and PARP under different combinations of transcribed data $D_T$ and untranscribed data $D_U$ for finetuning with semi-supervised distillation as described in Eq. 2. We fix the $D_T$ as LibriSpeech-1h and vary the resources in $D_U$. We adopt a NAT skip ratio of 0.4~0.8 and a sparsity ratio of 0.2~0.4 for our S$^6$-DAMON and a sparsity ratio of 0.1~0.5 for other ASR pruning baselines.

Observation and analysis. As shown in Fig. 3, we can observe that (1) our S$^6$-DAMON consistently outperforms PARP and OMP by a notable margin, e.g., an absolute >7% WER reduction as compared to PARP for achieving >64% FLOPs savings on wav2vec2-base with LibriSpeech-1h/100h as $D_T/D_U$; (2) our S$^6$-DAMON shows decent scalability under more stringent low-resource settings where PARP/OMP fail to achieve acceptable recognition effectiveness, e.g., an absolute up-to-34% lower WER over PARP when only LibriSpeech-1h is available; (3) our method can achieve a comparable WER (+0.5%) with a 40.8% reduction in FLOPs as compared to the original wav2vec2-base with

only 1h transcribed speech plus 100h untranscribed speech available; (4) enabling both SALAD and SAFARI can consistently win a better WER-FLOPs trade-off over $\overline{\text{SAFARI}}$, especially under more stringent low-resource settings, e.g., $S^6$-DAMON achieves an absolute 4.51% lower WER over SAFARI for reducing >64% FLOPs on LibriSpeech-1h. This indicates that input data sparsity is a critical dimension for structurally trimming down the complexity of speech SSL models.

**Benchmark under more resources.** As shown in Fig. 1 (b), given more downstream resources, e.g., the transcribed LibriSpeech-100h, our $S^6$-DAMON (1) still outperforms PARP and OMP, e.g., an absolute 10.96% lower WER for reducing >64% FLOPs on wav2vec2-base as compared to PARP, and (2) reduces 46% FLOPs of the original wav2vec2-base with a comparable WER (+0.57%).

**Benchmark under unstructured sparsity.** We further validate the scalability of our SAFARI to unstructured sparsity via benchmarking with the reported results of PARP, OMP, IMP, and MPI (i.e., magnitude pruning at pretrained initializations) in (Lai et al., 2021) on LibriSpeech-1h without any distillation. As shown in Fig. 4, we can observe that our SAFARI can still outperform all baseline methods, especially under large sparsity ratios, e.g., an absolute 8.61% lower WER under 80% sparsity over PARP. This indicates that (1) the sparsity discrepancy issue still exists in unstructured pruning under a larger sparsity ratio, and (2) our SAFARI pipeline consistently shows its superiority as an ASR pruning paradigm over PARP under both structured/unstructured sparsity patterns.

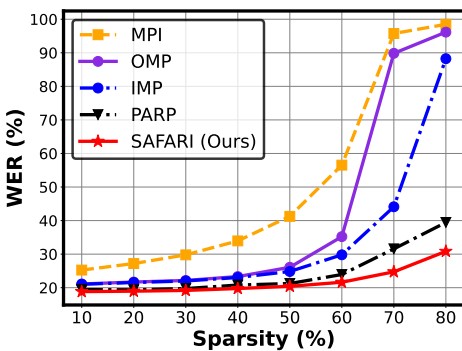

Figure 4: Benchmark our SAFARI with PARP, OMP, IMP, and MPI for unstructured pruning on wav2vec2-base/LibriSpeech-1h.

**Benchmark with distillation-based models.** We benchmark with the reported ASR results in Distil-HuBERT (Chang et al., 2022) and FitHuBERT (Lee et al., 2022) for compressing hubert and wav2vec2-base on LibriSpeech-100h. As shown in Tab. 2, $S^6$-DAMON achieves an absolute 7.04% lower WER with 8.7% fewer parameters as compared to the strongest baseline FitHuBERT, indicating that given a speech SSL model, trimming down its complexity

Table 2: Benchmark our $S^6$-DAMON with two distillation-based efficient ASR models.

| Method | Model | Params (M) | WER (%) |
|---|---|---|---|
| DistilHuBERT | hubert | 23.49 | 13.37 |
| FitHuBERT | wav2vec2 | 22.49 | 14.77 |
| | hubert | 22.49 | 12.66 |
| $S^6$-DAMON (Ours) | wav2vec2 | 20.53 | **7.73** |
| | hubert | 20.53 | **7.94** |

in a top-down manner may achieve better compression effectiveness than manually designing an efficient model from scratch without exploiting the intrinsic properties of human speech.

**Benchmark on more speech SSL models.** We further extend our $S^6$-DAMON to more models, i.e., data2vec (Baevski et al., 2022) and wav2vec2-large on LibriSpeech-1h (i.e., no $D_U$). As shown in Fig. 5, we can observe that (1) our method shows consistent effectiveness across models and starting from better speech SSL models like data2vec could lead to better WER-FLOPs trade-offs, and (2) according to the comparison between wav2vec2-base/large, structurally compressing a larger speech SSL model may not result in better WER-FLOPs trade-offs than compressing a smaller one as aggressively compressing a pretrained model could demolish the speech representation learned by the delicate SSL pretraining process. Therefore, a smaller model plus mild structured sparsity is favored.

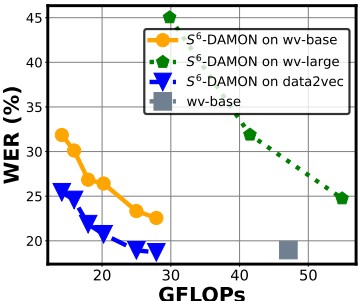

Figure 5: Apply $S^6$-DAMON on wav2vec2-base/large and data2vec.

### 4.3 EXTENSION TO OTHER SPEECH PROCESSING TASKS

**Benchmark on SUPERB.** Although efficient ASR is our main focus, we also evaluate our method on more speech processing tasks from SUPERB. In particular, we transfer the compressed models with LibriSpeech-1h/100h as $D_T/D_U$ of our $S^6$-DAMON and PARP to perform six speech processing tasks on SUPERB (Yang et al., 2021). As shown in Tab. 3, we observe that (1) our method wins four out of six tasks over the original wav2vec2-base with >=55.07%/44.39% FLOPs/parameter reductions, and (2) our method consistently outperforms PARP across all the tasks. This indicates that our $S^6$-DAMON can potentially serve as a general compression technique for speech processing.

Table 3: Benchmark our $S^6$-DAMON with PARP on six speech processing tasks from SUPERB.

| Method | FLOPs Saving (%) | Params Saving (%) | ER (Acc ↑) | KS (Acc ↑) | QbE (MaxTWV ↑) | IC (Acc ↑) | SF (F1 ↑) | ASV (EER ↓) |
|---|---|---|---|---|---|---|---|---|
| Original | 0.00 | 0.00 | 0.626 | 0.962 | 0.053 | 0.966 | **0.874** | **0.061** |
| PARP | 53.52 | 57.14 | 0.622 | 0.957 | 0.075 | 0.967 | 0.851 | 0.065 |
| $S^6$-DAMON (Ours) | 57.07 **69.98** | 44.39 **57.14** | **0.654** 0.641 | **0.964** 0.957 | 0.118 **0.122** | 0.982 **0.983** | 0.867 0.855 | 0.063 0.063 |

Table 4: Measure the latency of delivered models by our method and PARP on a Google Pixel 3 mobile phone. "FE" denotes the feature extractor and "Trans." denotes the transformer backbone. All models are finetuned on transcribed LibriSpeech-100h and "sr"/"sp" are the adopted skip ratio/sparsity.

| Method | sr / sp | Params (M) | WER (%) | Lat. (ms) Conv FE | Lat. (ms) Trans. | Speed-up on Trans. | Speed-up Overall | RTF ↓ |
|---|---|---|---|---|---|---|---|---|
| Original | - / - | 94.74 | 5.50 | 2270.1 | 6678.5 | 1.00× | 1.00× | 0.895 |
| PARP | - / 0.3 | 56.77 | 7.82 | 2270.1 | 4809.4 | 1.39× | 1.26× | 0.708 |
| | - / 0.4 | 45.80 | 11.69 | 2270.1 | 3756.6 | 1.77× | 1.48× | 0.603 |
| $S^6$-DAMON (Ours) | 0.6/0.2 | 69.01 | 6.07 | 2270.1 | 3834.5 | 1.74× | 1.47× | 0.610 |
| | 0.6/0.3 | 56.77 | 6.53 | 2270.1 | 3318.7 | 2.01× | 1.60× | 0.559 |
| | 0.8/0.3 | 56.77 | 6.98 | 2270.1 | 3061.1 | 2.18× | 1.68× | 0.533 |
| | 0.8/0.4 | 45.80 | 7.98 | 2270.1 | 2377.9 | 2.81× | 1.93× | 0.465 |
| | 0.8/0.7 | 20.53 | 9.20 | 2270.1 | 819.1 | 8.15× | 2.90× | 0.309 |

## 4.4 REAL-DEVICE MEASUREMENT OF $S^6$-DAMON

To validate the real-device efficiency of $S^6$-DAMON's delivered models, we measure their latency on a Google Pixel3 mobile phone for processing a 10s audio segment with a 16k sampling rate. We also report a real-time-factor (RTF) defined as the inference time divided by utterance duration (Gondi, 2022). As shown in Tab. 4, our $S^6$-DAMON can achieve (1) a 1.96× speed-up over PARP with an absolute 2.49% lower WER, and (2) a 1.60× speed-up over the original wav2vec2-base with a comparable WER (+1.03%) or a 2.90× speed-up while maintaining the absolute WER within 10%. This indicates that our method can outperform SOTA ASR pruning methods in real-device efficiency and significantly bridge the gap between speech SSL models and real-time speech recognition.

## 4.5 ABLATION STUDIES

**Benchmark with random, uniform, and other adaptive token skipping methods.** To validate the skipping strategies made by SALAD, we benchmark with three token skipping methods: (1) random skip, (2) uniform skip with a similar effect as reducing the sampling rate, and (3) layer-wise adaptive skip (Wang et al., 2021) based on attention scores, which is originally designed for NLP. We finetune wav2vec2-base with each of the skipping methods on LibriSpeech-1h. We control their skip ratios to ensure a comparable FLOPs saving. As shown in Tab. 5, our SALAD consistently wins the lowest WER under comparable FLOPs and the adaptive skip method can hardly surpass the simple uniform skip strategy, indicating that without considering the intrinsic properties of human speech, the monotonic alignment between input speech and output transcriptions can be easily demolished.

Table 5: Benchmark with different token skipping strategies.

| Method | FLOPs Saving (%) | WER (%) |
|---|---|---|
| **SALAD** | **24.3** **32.0** | **19.17** **20.21** |
| Uniform Skip | 20.0 30.0 | 19.89 24.63 |
| Random Skip | 20.0 30.0 | 43.98 79.87 |
| Adaptive Skip | 15.0 22.5 | 25.658 36.193 |

## 5 CONCLUSION

Both the lack of large-scale transcribed speech data for low-resource spoken languages and the prohibitive model complexity hinder ubiquitous DNN-powered ASR systems on mobile platforms. This work proposes $S^6$-DAMON to tackle both challenges via effectively pruning speech SSL models to enable real-time on-device ASR. Specifically, $S^6$-DAMON integrates SALAD and SAFARI to pursue structured sparsity in both input data and model structures, respectively, where the former exploits and intrinsic properties of human speech and the latter reduces the sparsity discrepancy between finetuning/deployment and enhances the learnability of sparsity distributions. $S^6$-DAMON has enabled the deployment of speech SSL models on mobile devices based on our extensive experiments and can shed light on future innovations on efficiency-oriented speech SSL paradigms.

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

# A   MORE ABLATION STUDIES OF S$^6$-DAMON

## A.1   MODULE-WISE SENSITIVITY TO STRUCTURED PRUNING

We apply SAFARI on wav2vec2-base on top of LibriSpeech-1h and vary the sparsity in FFN and SA under a comparable FLOPs . As shown in Tab. 6, we consistently find that FFNs are more sensitive to structured pruning especially under large sparsity, which may be because task-specific information is mostly learned by FFNs during finetuning thus their sufficient complexity is crucial. Therefore, we by default set their sparsity to satisfy $(sp_{SA} + sp_{FFN})/2 = sp$ and $sp_{SA} - sp_{FFN} = 0.2$ for a given sparsity $sp$ in Sec. 4.

Table 6: Benchmark the WERs with varied FFN/SA sparsity on wav2vec2-base/LibriSpeech-1h.

| FFN/SA sparsity | GFLOPs | WER (%) | FFN/SA sparsity | GFLOPs | WER (%) | FFN/SA sparsity | GFLOPs | WER (%) |
|---|---|---|---|---|---|---|---|---|
| 0.2/0.2 | 33.024 | 23.19 | 0.3/0.3 | 26.96 | 23.19 | 0.4/0.4 | 21.52 | 31.97 |
| 0.1/0.3 | 33.96 | **22.13** | 0.2/0.4 | 27.61 | **24.52** | 0.3/0.5 | 21.89 | **29.72** |
| 0.3/0.1 | 32.016 | 25.19 | 0.4/0.2 | 26.23 | 31.05 | 0.5/0.3 | 21.08 | 47.77 |

## A.2   THE CHOICE OF ADJUSTMENT RATIOS

We vary the adjustment ratio $ar$ under different sparsity $sp$ on top of wav2vec2-base and LibriSpeech-1h. As shown in Tab. 7, we observe that the optimal $ar$ varies for different $sp$ and in general larger sparsity calls for higher learnability of sparsity distributions. Therefore, we set $ar = sp$ by default in Sec. 4.

Table 7: Benchmark the WERs with varied adjustment ratios $ar$ under different sparsity $sp$.

| $ar$ / $sp$ | 0.2 | 0.3 | 0.4 |
|---|---|---|---|
| 0.1 | 23.75 | 26.53 | 33.14 |
| 0.2 | **22.86** | 28.65 | 31.97 |
| 0.3 | 22.96 | **23.19** | 30.76 |
| 0.4 | 23.18 | 26.79 | **30.74** |

## A.3   THE NECESSITY OF ENFORCING HIGH RECALL ON SATs

To validate this, we train two SALADs w/ and w/o recall-aware training (RAT), which are next applied on wav2vec2-base with different NAT $sr$ on LibriSpeech-1h. As shown in Tab. 8, explicitly enforcing a high recall on SATs results in consistent lower WER especially under larger $sr$, validating the necessity of maximally covering all SATs.

Table 8: Validate the necessity of RAT.

| Setting | w/o RAT | w/ RAT |
|---|---|---|
| Acc (%) | **79.38** | 75.69 |
| Recall (%) | 64.38 | **89.08** |
| NAT $sr$=0.4 | 19.56 | **18.42** |
| NAT $sr$=0.6 | 21.63 | **19.17** |
| NAT $sr$=0.8 | 23.89 | **20.21** |

## A.4   CROSS-LINGUAL TRANSFER OF SALAD

Considering the semi-supervised training scheme of SALAD requires a large set of untranscribed speech, which may not be available for some spoken languages, we evaluate whether the SALAD trained on English can be directly transferred to detect SATs/NATs for other languages. In particular, we transfer SALAD trained on untranscribed LibriSpeech-100h to pursue the input sparsity for finetuning wav2vec2-base on Dutch, Spanish, and Mandarin from CommonVoice (Ardila et al., 2019). As shown in Tab. 9, we can see that SALAD trained on English can transfer well to other languages, e.g., achieve a comparable or

Table 9: Evaluate the phoneme recognition rate (PER) when applying SALAD trained on English to other languages under different $sr$.

| $sr$ | Dutch | Spanish | Mandarin |
|---|---|---|---|
| - | 19.82 | 13.86 | 26.67 |
| 0.2 | 18.89 | 13.76 | 26.61 |
| 0.4 | 19.16 | 13.85 | 26.89 |
| 0.6 | 19.55 | 13.99 | 26.84 |
| 0.8 | 20.09 | 14.32 | 28.46 |

lower PER under 0.6 $sr$, indicating that SALAD can extract general phonetic features that can be shared across spoken languages.

## B   MORE DETAILS ABOUT EXPERIMENT SETUP

**Finetuning settings.** We implement $S^6$-DAMON on top of fairseq (Ott et al., 2019) and we follow the default finetuning settings for each task, i.e., the default configurations in fairseq for ASR/PR and those in SUPERB (Yang et al., 2021) for other speech processing tasks. In particular, all experiments on ASR/PR are trained for 12k/15k/20k/80k steps on the 10m/1h/10h/100h splits using an Adam optimizer with an initial learning rate of 5e-5 plus a tri-stage schedule (Baevski et al., 2020). We do not freeze all the transformer layers for the first 10k steps (Baevski et al., 2020), following (Lai et al., 2021). In addition, considering the transformer backbone accounts for >90% parameters in the speech SSL models, all the reported FLOPs/Params savings and sparsity ratios are relative to the transformer, following PARP (Lai et al., 2021).

**Measurement settings.** For the measurement on the Google Pixel 3 mobile phone, all Pytorch models are converted to ONNX and then compiled to the TFLite format, following (Li et al., 2021a). We separately compile (a) the convolutional feature extractor + SALAD, and (b) the transformer backbone, where the output of (a) is fed into (b) as its input. The latency on both (a) and (b) as well as the overall speed-up are reported in Sec. 4.4.

