# OpenReview forum: "S$^6$-DAMON: Bridging Self-Supervised Speech Models and Real-time Speech Recognition"
_ICLR.cc/2023/Conference — Submitted to ICLR 2023_

### Official Review · Reviewer_V9mU · 2022-10-21

**Confidence:** 4
**Correctness:** 3
**Technical Novelty And Significance:** 3
**Empirical Novelty And Significance:** 2
**Recommendation:** 5

**Clarity, Quality, Novelty And Reproducibility:**

# Clarity

The paper is mostly clearly written. I found that the section 3.4 could be extended with more details about the center model of the paper. For example, I did not understand how the argmax is computed (examples are sampled or really summed over the whole set?). There are a number of small issues with writing such as repeating expressions. I also found it very distracting to use such abbreviations for model names -- this is just over the top.

# Quality

Generally, the paper has above average quality. The methods are sound and and there is an attempt to experimentally test all the claims. One of the highlights of the paper I liked, and I don't see frequently, is testing on a real device. As mentioned before, the paper lacks focus by introducing several disconnected methods. Then, the presentation of the results needs alignment with the prior works as the faulty baseline might diminish the support for the claims.

# Novelty

Each proposed method has average novelty, but the combination of them seems to be novel.

# Reproducibility

The paper introduces several methods with many hyperparameters. The hyperparameters are given in the paper, but it might be challenging to reproduce all the implementation detals. The paper does not provide error bars and doesn't test the models for stability and robustness.

# Other

There is a recurring mistake in the paper: difference in percentages is not a percentage https://en.wikipedia.org/wiki/Percentage_point

**Strength And Weaknesses:**

Strengths:
  - The paper tackles a challenging problem.
  - The proposed models are sound. The design makes sense and motivated well
  - The paper provides evaluations on several datasets: Librispeech and Suberb
  - The paper evaluates the latency on a real device

Weaknesses:
  - There is some lack of focus in this paper: there are three models introduced. Each one is complex enough to have a small paper. This leads to a very condensed writing and each model is not described well.
  - The results are hard to compare to previous works. For example, the Fig. 3 models are trained in a different setup than wave2vec2 or HuBERT, therefore not comparable. The PARP paper's setup is also a bit different than presented here.

**Summary Of The Paper:**

This paper tackles the problem of on-device ASR self-supervised learning. In this scenario, there is a large unlabeled dataset is available for unsupervised pre-training, and a much smaller supervised set for fine-tuning. Previous models such as HuBERT, data2vec use a large transformer model for the feature pre-training. Such a transformer is resource hungry for the scenario of running on an mobile phone or an embedded device.

This work proposes 3 different components. First, there is a method to prune the input tokens. The paper uses a teacher ASR model to train a small classifier which predicts which input token is salient. Examples of salient tokens are the first of the repeating letters, or the first of several silences.

Second proposed component is the method to sparcify the model. The paper builds upon the PARP and proposes an iterative procedure of sparcification and finetuning steps.

Finally, the third proposed model combines both previous models to use them for semi-supervised distillation from a teacher model.

The paper reports the results on Librispeech and Superb corpora and then evaluates the latency on a real device.

**Summary Of The Review:**

Given the previous assessment, now I recommend reject.

But this paper has a potential, and here are the ways to improve it:
  - Align the experimental results to the prior literature: HuBERT, data2vec, wav2vec, PARP. If it is not possible, provide a disclaimer (for example, that the baseline was trained on a larger corpus and therefore had better performance).
  - Tie the different components together.

---

> ### Author Response · Authors · 2022-11-19
> **Response to Reviewer V9mU (Part 1)**
>
> Thanks for recognizing the importance of our work and the value of our real-device measurements. We have addressed your comments/concerns as follows:
>
> **1. Lack of focus in the presented techniques**
>
> Thanks for pointing this out! We humbly clarify that our framework features a data-model co-compression method for speech SSL models, where the two presented techniques (SALAD and SAFARI) trim down the redundancy on the data and model level, respectively, and thus they are not disconnected. According to the evaluation in Figure 3 of our manuscript, enabling both data and model compression leads to the best accuracy and efficiency trade-off, indicating that both of them non-trivially contribute to the advance towards the on-device deployment of speech SSL models. We will follow your suggestion to better organize the presentation to improve the presentation clarity in the final version.
>
> **2. Clarifications about the evaluation settings**
>
> **We kindly clarify that we have adopted both the standard evaluation settings following wav2vec2[1]/PARP[2] and the extended evaluation settings following Distilhubert[3]/FitHuBERT[4]  to ensure that the benchmark with SOTA baselines is as fair as possible.**
>
> The key difference between the standard and extended evaluation settings is the existence of D$_U$, i.e., the unlabeled data, which is utilized by the distillation term in Eq.(2) of our manuscript to boost the compression effectiveness according to Distilhubert[3]/FitHuBERT[4]. We have benchmarked with both the baselines w/ and w/o the existence of D$_U$. In particular, the experiments in Figure 1(b) (i.e., 100h labeled speech), Figure 3(a) (i.e., 1h labeled speech), and Figure 4 (i.e., 1h labeled speech with an unstructured sparsity) all strictly follow the finetuning settings in wav2vec2 and PARP w/o D$_U$, where the details are provided in Appendix B. In addition, to further enhance the achievable accuracy-efficiency trade-off of both our method and the baselines, we enable the distillation w/ D$_U$ in the experiments shown in  Figure 3(b)/3(c) (i.e., 1h labeled speech with 10h/100h unlabeled speech, respectively) and other ablation study figures/tables. The adopted settings are elaborated in the text of our submitted manuscript.
>
> Furthermore, we have added an additional benchmark in the table below to cover more low-resource settings (D$_T$=Libri-10m and Libri-10h) adopted by wav2vec2/PARP (with D$_U$=Libri-100h to improve the achievable accuracy). We can observe that our S$^6$-DAMON can consistently outperform all the baselines under different resources of labeled speech, e.g., an absolute 6.64%/13.34% WER reduction over PARP/OMP, when achieving a >60% FLOPs saving given 10m labeled speech, indicating the decent scalability of our S$^6$-DAMON across different resource settings.
>
> |  | S$^6$-DAMON |  |  | PARP |  |  | OMP |  |
> |---|:---:|---|---|:---:|---|---|:---:|---|
> | FLOPs Savings | Libri-10m | Libri-10h | FLOPs Savings | Libri-10m | Libri-10h | FLOPs Savings | Libri-10m | Libri-10h |
> | 40.78% | **35.98** | **9.72** | 27.90% | 37.12 | 10.03 | 27.90% | 39.32 | 10.37 |
> | 46.94% | **36.59** | **11.08** | 41.38% | 38.93 | 13.46 | 41.38% | 43.63 | 14.54 |
> | 57.07% | **36.75** | **11.71** | 53.52% | 41.71 | 17.33 | 53.52% | 46.27 | 20.33 |
> | 66.18% | **37.78** | **14.76** | 64.29% | 44.42 | 20.87 | 64.29% | 50.12 | 27.69 |
>
> We believe that we have effectively clarified this point, and please let us know if you have any further questions about the evaluation settings.
>
> **3. Reproducibility**
>
> We humbly note that the detailed experimental setup is provided in Section 4.1 and Appendix B of our submitted manuscript. Furthermore, as promised in the abstract of our submitted manuscript, all our source codes and trained models will be released upon acceptance.

---

> > ### Author Response · Authors · 2022-11-19
> > **Response to Reviewer V9mU (Part 2)**
> >
> > **4. Clarifications about the details in Section 3.4**
> >
> > For updating $\theta$ in Eq.(2) of Section 3.4, we follow the standard finetuning settings in wav2vec2[1] to update $\theta$ based on each sampled batch instead of the whole dataset. We will include more details and improve writing clarity in the final version.
> >
> >
> > **5. Abbreviations for technique names**
> >
> > Thanks for pointing this out! Our intention is to use abbreviations for reducing reading redundancy and improving the clarity. To follow your suggestion and avoid the confusion, we will consider rephrasing the names of our techniques in the final version.
> >
> > **6. The mistake for presenting percentages**
> >
> > Thanks for pointing this out! We have corrected this issue via using an absolute WER reduction, following in PARP[2], in our revised manuscript.
> >
> > **Reference:**
> >
> > [1] “wav2vec 2.0: A Framework for Self-Supervised Learning of Speech Representations”, A. Baevski et al., NeurIPS 2020.
> >
> > [2] “PARP: Prune, Adjust and Re-Prune for Self-Supervised Speech Recognition”, C. Lai et al., NeurIPS 2021.
> >
> > [3] “Distilhubert: Speech Representation Learning by Layer-Wise Distillation of Hidden-Unit Bert”, H. Chang et al., ICASSP 2022.
> >
> > [4] “FitHuBERT: Going Thinner and Deeper for Knowledge Distillation of Speech Self-Supervised Learning”, Y. Lee et al., InterSpeech 2022.

---

### Official Review · Reviewer_JZjt · 2022-10-24

**Confidence:** 4
**Correctness:** 3
**Technical Novelty And Significance:** 2
**Empirical Novelty And Significance:** 2
**Recommendation:** 5

**Clarity, Quality, Novelty And Reproducibility:**

The paper is clearly written. Novelty is not enough. No code or trained-model released for reproduction.

**Strength And Weaknesses:**

Pros:
1. The paper is clearly written and easy to follow. The illustrative figures and examples are helpful to understand the ideas.
2. Examples are conducted on different backbone models including wav2vec2 and data2vec, with different sizes.
3. The ablation study compared with random removing frames or adaptive removing, are very useful to establish a sense of how the proposed method performs compared with baselines.
4. The pruning performance is impressive compared with other methods. I guess it is due to the token skipping

Cons:
1. The idea of both methods are incremental, though combining them and show they works is the value of this paper.
2. In figure 3 (a), why the S6-DAMON does not have values below 40% FLOPs? Is it that the WER is worse than the others?

**Summary Of The Paper:**

This paper proposes two methods for efficient ASR modeling, the token/frame skipping and structured pruning. The token skipping is achieved by applying a binary classifier to detect whether a token is salient. The structured pruning is an iterative process of sparsifying, fine-tuning and recovering, until reaching a certain sparsity level. Experimental results demonstrates the effectiveness in terms of FLOPs savings against WER increases, in comparison with other popular pruning methods.


**Summary Of The Review:**

Overall, though the papers' core ideas are not novel, the combination and comprehensive experimental evaluation show that it is working well. One concern is that no source code nor trained model provided for reproduction.

---

> ### Author Response · Authors · 2022-11-19
> **Response to Reviewer JZjt**
>
> Thanks for recognizing the comprehensiveness of our experimental evaluation. We have addressed your comments/concerns as follows:
>
> **1. Novelty of our method**
>
> We humbly clarify that our work has made non-trivial contributions in the emerging field of SSL model compression from the following aspects:
>
> *(1)* As emphasized in the introduction section of our manuscript, **compressing SSL pretrained models is non-trivially different from commonly studied supervised model compression**. This is because it is difficult to learn a sparsity distribution during finetuning while at the same time maintaining the fidelity of the pretrained representation due to the low-resource downstream data. It is worth noting that our work is among the early works to identify, analyze, and tackle the failures when applying commonly used compression techniques to compress SSL models (e.g., in Section 3.3 of our manuscript) and our delivered insights can also shed light on future innovations along this emerging and important field.
>
>
> *(2)* Our work is **the first to enforce a structured sparsity constraint in speech SSL models for their deployment on real devices via data-model co-compression**: on the data side, we are the first to leverage the redundancy in repeated/blank regions of human utterances to compress speech SSL models’ input tokens; on the model size, we identify and analyze the issues of SOTA ASR pruning methods and propose a new pruning pipeline.
>
> *(3)* Both of our proposed techniques mentioned above have been validated to be able to **significantly and consistently improve the accuracy-efficiency trade-off over SOTA methods by extensive experiments and on-device measurements**. As the first work that targets real-device deployment of speech SSL models, we can expect that our framework can serve as both a cornerstone and a strong baseline in this direction and inspire future innovations to further push forward the frontier.
>
> **2. Clarifications about the FLOPs Savings in Figure 3(a)**
>
> We kindly clarify that the x-axis in Figure 3 is FLOPs Savings (i.e., the larger value indicates better efficiency) and thus all data points of S$^6$-DAMON have achieved >=40% FLOPs reduction. In addition, SAFARI in Figure 3 is also our proposed method (which only considers model compression, i.e., integrating our proposed model compression technique) and has already outperformed all the baselines. S$^6$-DAMON integrates both SAFARI and SALAD for enabling data-model co-optimization to further push forward the achievable accuracy-efficiency trade-off.
>
> Furthermore, following your suggestion, we also provide the data points with <40% FLOPs Savings (i.e., worse efficiency but lower WER) in the table below, which follows the same setting as Figure 3(a) in our manuscript (D$_T$=Libri-1h/D$_U$=None). We can observe that our S$^6$-DAMON consistently achieves the best WER-FLOPs trade-off, indicating that enabling both our SAFARI and SALAD techniques to exploit both model and data redundancy is a more effective compression strategy.
>
> | S$^6$-DAMON (Ours) |  | SAFARI (Ours) |  | PARP |  | OMP |  |
> |:---:|---|:---:|---|:---:|---|---|---|
> | FLOP Savings | WER (%) | FLOPs Savings | WER (%) | FLOPs Savings | WER (%) | FLOPs Savings | WER (%) |
> | 15.60% | **19.37** | 13.00% | 19.99 | 13.00% | 21.22 | 13.00% | 24.47 |
> | 30.51% | **20.13** | 27.90% | 22.14 | 27.90% | 23.71 | 27.90% | 30.10 |
> | 40.78% | **22.56** | 41.38% | 26.17 | 41.38% | 34.57 | 41.38% | 58.12 |
> | 57.07% | **26.42** | 53.52% | 29.48 | 53.52% | 50.68 | 53.52% | 78.18 |
> | 66.18% | **30.12** | 64.29% | 34.63 | 64.29% | 64.52 | 64.29% | 87.39 |
>
>
> **3. Source codes and trained models**
>
> As promised in the abstract of our submitted manuscript,  all our source codes and trained models will be released upon acceptance.

---

### Official Review · Reviewer_jv1R · 2022-10-25

**Confidence:** 4
**Correctness:** 4
**Technical Novelty And Significance:** 3
**Empirical Novelty And Significance:** 2
**Recommendation:** 5

**Clarity, Quality, Novelty And Reproducibility:**

The article is clear. The contribution mostly is in the performance improvement.

**Strength And Weaknesses:**

Strength: the results in the paper show significant improvement compared to the previous work.
Weakness:
(1) The comparisons with the previous approach are limited, making it less convincing.
(2) The comparisons with other pruning algorithms are also lacking.
(3) SALAD is only slightly better than a simple uniform skipping.
(4) The proposed pruning algorithm is a minor extension of PARP, making its novelty limited.


**Summary Of The Paper:**

This paper applies two approaches for improving the speed of ASR models: learning to identify salient tokens and skip those that are identified not salient, and weight pruning. The results show some improvements on the SUPERB set and LibriSpeech-100h. The authors also benchmark the inference speed on Pixel 3 phone and demonstrate good speedup with the technique.
Two questions:
(1) in section 4.5, when comparing different skipping methods, are the models also fine-tuned with those skipping methods? Or are they only applied during inference?
(2) On the SUPERB set, why would the quality be better than the uncompressed model?

A more recent related work on self-supervised learning for speech is missing:
Self-supervised learning with random-projection quantizer for speech recognition. ICML 2022

**Summary Of The Review:**

The proposed approach shows good improvement compared with PARP, but the empirical studies are a bit limited. There should also be more comparisons with other algorithms. The technical novelty is generally limited.

---

> ### Author Response · Authors · 2022-11-19
> **Response to Reviewer jv1R (Part 1)**
>
> Thanks for recognizing the significant experimental improvements of our method. We have addressed your comments/concerns as follows:
>
> **1. Technical novelty of our work and the difference over PARP**
>
> We humbly clarify that our work has made non-trivial contributions in the emerging field of SSL model compression from the following aspects:
>
> *(1)* As emphasized in the introduction section of our manuscript, **compressing SSL pretrained models is non-trivially different from commonly studied supervised model compression**. This is because it is difficult to learn a sparsity distribution during finetuning while at the same time maintaining the fidelity of the pretrained representation due to the low-resource downstream data. It is worth noting that our work is among the early works to identify, analyze, and tackle the failures when applying commonly used compression techniques to compress SSL models (e.g., in Section 3.3 of our manuscript) and our delivered insights can also shed light on future innovations along this emerging and important field.
>
> *(2)* Our work is **the first to enforce a structured sparsity constraint on speech SSL models for their deployment on real devices via data-model co-compression**: on the data side, we are the first to leverage the redundancy in repeated/blank regions of human utterances to compress speech SSL models’ input tokens, which outperforms uniform skipping as elaborated in our response to Q3 below; on the model size, we identify and analyze the issues of SOTA ASR pruning methods and propose a new pruning pipeline, which is non-trivially different from PARP: PARP is essentially an iterative pruning method that allow the pruned weights to be trained during finetuning; In contrast, our method features a completely different pipeline, where the pruned weights are enforced to be zero during finetuning and the sparsity distribution is adjusted via a new stage to grow back the neurons that contribute the most to the finetuning process based on their gradient magnitudes. More details are discussed in the implementation of SAFARI in Section 3.3 of our manuscript.
>
>
> *(3)* Both of our proposed techniques mentioned above have been validated to be able to **significantly and consistently improve the accuracy-efficiency trade-off over SOTA methods by extensive experiments and on-device measurements**. As the first work that targets real-device deployment of speech SSL models, we can expect that our framework can serve as both a cornerstone and a strong baseline in this direction and inspire future innovations to further push forward the frontier.
>
>
> **2. The comparison with previous approaches and pruning algorithms**
>
> We humbly clarify that we have benchmarked with all the existing methods that target to improve efficiency of speech SSL models on ASR tasks, including a pruning-based method PARP[1] and two knowledge distillation methods [2][3], across both structured/unstructured settings in Figure 3/4 and Table 2 of our submitted manuscript. In addition, as ASR pruning is still in its infancy, we have also included the existing ASR pruning methods based on the reported results from PARP[1] as our baselines. Furthermore, extensive ablation studies that have evaluated and justified our design choices are provided in both the main text and the appendix of our submitted manuscript. Finally, our provided experiments are recognized by Reviewer JZjt as “comprehensive experimental evaluation” and our real-device measurement is recognized by Reviewer V9mU as “a highlight”.
>
> In addition, to further address your concern, we have also added the benchmark with PARP/OMP under different resources (D$_T$=Libri-10m/Libri-10h with D$_U$=Libri-100h) as a complement to Figure 3 of our submitted manuscript in the table below. We can observe that our S$^6$-DAMON can consistently outperform all the baselines under different resources of labeled speech, e.g., an absolute 6.64%/13.34% WER reduction over PARP/OMP, respectively, when achieving a >60% FLOPs saving given 10m labeled speech, indicating the decent scalability of our S$^6$-DAMON across different resource settings.
>
> |  | S$^6$-DAMON |  |  | PARP |  |  | OMP |  |
> |---|:---:|---|---|:---:|---|---|:---:|---|
> | FLOPs Savings | Libri-10m | Libri-10h | FLOPs Savings | Libri-10m | Libri-10h | FLOPs Savings | Libri-10m | Libri-10h |
> | 40.78% | **35.98** | **9.72** | 27.90% | 37.12 | 10.03 | 27.90% | 39.32 | 10.37 |
> | 46.94% | **36.59** | **11.08** | 41.38% | 38.93 | 13.46 | 41.38% | 43.63 | 14.54 |
> | 57.07% | **36.75** | **11.71** | 53.52% | 41.71 | 17.33 | 53.52% | 46.27 | 20.33 |
> | 66.18% | **37.78** | **14.76** | 64.29% | 44.42 | 20.87 | 64.29% | 50.12 | 27.69 |
>
> Lastly, if you could kindly recommend other ASR compression approaches or pruning algorithms that can serve as our baselines, we will be happy to add additional benchmark with them during the next-run response or in the final version.

---

> > ### Author Response · Authors · 2022-11-19
> > **Response to Reviewer jv1R (Part 2)**
> >
> > **3. Clarification about SALAD’s decent improvements over uniform skipping**
> >
> > We kindly clarify that SALAD can achieve notable improvements over uniform skipping, e.g., under a token skip ratio of 30%, SALAD can achieve an absolute 4.42% WER reduction over uniform skipping according to Table 5 of our submitted manuscript.
> >
> > In addition, we further clarify that under a small token skip ratio (<=20%) where most information of the input speech signals are maintained, uniform skipping can achieve satisfactory effectiveness, e.g., slightly worse performance (+0.72% WER) over our SALAD technique; On the other hand, under larger token skip ratios, SALAD outperforms uniform skipping by a notable margin, e.g., an absolute 15.90% WER reduction under a skipping ratio of 50% as shown in the table below, which follows the setting (wav2vec2-base on LibriSpeech-1h) in Table 5 of our submitted manuscript. We will further clarify this in the final version.
> >
> > | SALAD (Ours) |  | Uniform Skipping |  |
> > |:---:|---|:---:|---|
> > | FLOPs Savings | WER (%) | FLOPs Savings | WER (%) |
> > | 12.1% | **18.96** | 10% | 19.23 |
> > | 24.3% | **19.17** | 20% | 19.89 |
> > | 32.0% | **20.21** | 30% | 24.63 |
> > | 41.20% | **22.47** | 40% | 30.93 |
> > | 50.10% | **26.97** | 50% | 42.87 |
> >
> >
> > **4. Clarification about experiment settings in Section 4.5**
> >
> > For the ablation study in Section 4.5, we have finetuned the models with those skipping methods to maximize their achievable accuracy for a fair comparison, as clarified in the revised manuscript.
> >
> > **5. The better quality on SUPERB datasets**
> >
> > Thanks for this good question! We have observed that properly induced sparsity on speech SSL models can improve the accuracy on specific speech processing tasks, e.g., our S$^6$-DAMON has outperformed the best SOTA method on the QbE (Query by Example Spoken Term Detection) task with a 0.122 MTWV, where the previous SOTA on SUPERB leaderboard (achieved by WavLM Base+) is 0.0988 MTWV. We hypothesize that this is because most speech processing tasks rely on only a subset of the features encoded by speech SSL models, e.g., the features of key words, and thus properly induced sparsity can alleviate overfitting and help the model learn the most relevant features of the target task during finetuning. We will add such discussions in our final version.
> >
> > **6. The suggested related work**
> >
> > Thanks for your suggestion! We have cited this paper in our revised manuscript, and will consider applying our method on top of their provided speech SSL models once they open-source their codes.
> >
> > **Reference:**
> >
> > [1] “PARP: Prune, Adjust and Re-Prune for Self-Supervised Speech Recognition”, C. Lai et al., NeurIPS 2021.
> >
> > [2] “Distilhubert: Speech Representation Learning by Layer-Wise Distillation of Hidden-Unit Bert”, H. Chang et al., ICASSP 2022.
> >
> > [3] “FitHuBERT: Going Thinner and Deeper for Knowledge Distillation of Speech Self-Supervised Learning”, Y. Lee et al., InterSpeech 2022.

---

### Decision · Program_Chairs · 2023-01-20

**Decision:**

Reject

**Justification For Why Not Higher Score:**

All reviewers do not think the paper has sufficient novelty and think the experimental design can be better.

**Justification For Why Not Lower Score:**

N/A

**Metareview: Summary, Strengths And Weaknesses:**

The paper proposes two improvements, fine-tuning self-supervised models for automatic speech recognition. One is an additional classifier to remove frames, and the other is an modified iterative fine-tuning procedure. No hypotheses are given in the paper. Experiments show that the proposed approach can achieve better word error rates at lower sparsity level.

Two reviewers comment that the novelty is limited. Two reviewers mention that there are no comparison to prior work. One reviewer even points out that the improvements are limited.

Overall, the reviewers are lukewarm about the paper. Though all reviewers think the paper is clear, I think the paper would benefit from better writing. For example, there are too many subsubsections, and the paper feels fragmented. The related work is a laundry list of papers, and does not provide contrast between the paper and the prior work. The main message is buried in the experimental results. One reviewer mention a lack of focus, and that's the typical sign that the writing can be improved. The paper would be stronger if the writing can be improved.